# Protease Profile of Tumor-Associated Mast Cells in Melanoma

**DOI:** 10.3390/ijms23168930

**Published:** 2022-08-11

**Authors:** Dmitri Atiakshin, Andrey Kostin, Igor Buchwalow, Vera Samoilova, Markus Tiemann

**Affiliations:** 1Research and Educational Resource Centre for Immunophenotyping, Digital Spatial Profiling and Ultrastructural Analysis Innovative Technologies, Peoples’ Friendship University of Russia, 117198 Moscow, Russia; 2Research Institute of Experimental Biology and Medicine, Burdenko Voronezh State Medical University, 394036 Voronezh, Russia; 3Institute for Hematopathology, 22547 Hamburg, Germany

**Keywords:** mast cells, melanoma, tryptase, chymase, carboxypeptidases

## Abstract

Mast cells (MCs) produce a variety of mediators, including proteases—tryptase, chymase, and carboxypeptidases—which are important for the immune response. However, a detailed assessment of the mechanisms of biogenesis and excretion of proteases in melanoma has yet to be carried out. In this study, we present data on phenotype and secretory pathways of proteases in MCs in the course of melanoma. The development of melanoma was found to be accompanied by the appearance in the tumor-associated MC population of several pools with a predominant content of one or two specific proteases with a low content or complete absence of others. Elucidation of the molecular and morphological features of the expression of MC proteases in melanoma allows us a fresh perspective of the pathogenesis of the disease, and can be used to clarify MCs classification, the disease prognosis, and evaluate the effectiveness of ongoing antitumor therapy.

## 1. Introduction

Melanoma is the most fatal skin cancer. As a result, melanoma continues to be the subject of several preclinical and clinical investigations intended to further our understanding of cancer immunobiology [1]. Human and animal studies addressing potential functions of mast cells (MCs) in skin cancers have provided conflicting results. Many unanswered questions need to be addressed before we understand whether tumor-associated MC are adversaries, allies, or simply innocent bystanders in different types and subtypes of skin cancers [2].

MCs compose roughly 10% of all immunocompetent cells in the skin [3] and may be the first of the immunocompetent cells to migrate to tumor cells and take part in the formation of the biological properties of the tumor microenvironment in the course of tumor development [4,5]. Therefore, MCs apparently play opposing roles in tumor biology; the microenvironment could polarize MCs to possess either tumor-promoting or inhibitory effects. According to some authors, their accumulation in the tumor microenvironment may be associated with poor prognosis in cutaneous malignancies such as melanoma [6,7]. Contrarily, an increase in MCs directly in the tumor is considered by other authors as a favorable prognostic criterion for patients with melanoma [8,9]. MCs may also have an important role in inflammatory disease (such as psoriasis) and this may have also important retracements in the future regarding the immune response to melanoma cells, above all in patients under immunotherapy [10].

A possible pathogenetic linkage between cutaneous mastocytosis and melanoma is being actively discussed [11]; MCs appear to be involved either in antitumor processes or, conversely, in the mechanisms of oncogenesis stimulation. MC can utilize immunotropic mechanisms and induce antiproliferative effects to inhibit tumor growth [2,7,12,13,14,15,16]. On the other hand, MCs may contribute to the tumorigenesis of aggressive human cancers via versatile mechanisms, including extracellular matrix degradation. Through their specific proteases [17,18,19,20], MCs are involved in degradation of the matrix, which is required for tumor spread [16,21,22,23,24,25,26].

The physiological effects of tryptase, chymase, and carboxypeptidases allow them to be considered promising targets for the treatment of many diseases, including cancer [27,28]. Simulation of specific protease deficiency in the MCs in laboratory animals led to a higher degree of melanoma dissemination in the lungs [29,30]. In mice, the enzyme Mcpt6 (a human tryptase analogue) caused a decrease in melanocyte proliferation in vitro [29].

Studies on the biological effects of tryptase on the DNA state of tumor cells also evidence the antiproliferative effects of the MC proteases [29,31,32]. Low numbers of tryptase- and chymase-positive mast cells are associated with reduced survival and advanced tumor stage in melanoma [8,9,33]. Similarly, invasive melanomas were characterized by a lower content of tryptase- and chymase-positive MCs compared to melanomas in situ, or to benign and dysplastic nevi [8]. Furthermore, the protective role of MCs was supported by a study in which the artificially induced secretion of all MC mediators had an antiproliferative effect on cultured melanoma cells [9].

Overall, most findings evidence a protective role for MCs and their proteinases in the pathogenesis of melanoma ([8] and citations therein). However, a detailed study of the landscape of specific proteases of tumor-associated MCs using multiplex technology for the simultaneous detection of tryptase, chymase, and carboxypeptidases has not been performed yet. Therefore, the aim of this study was to elucidate the biogenesis and secretory pathways of specific proteases in intratumoral MCs in human melanoma.

## 2. Results

In this study, we performed multiple immunolabeling of tryptase, chymase, and carboxypeptidase A3 in patients with melanoma of the skin. The MC phenotypes in the intratumoral population were assessed basing on the content of tryptase, chymase, and carboxypeptidase A3 (CPA3). MC histotopography in the tumor microenvironment was assessed based on the localization of MCs in the immediate vicinity or in the area of paracrine influence of tumor cells, leukocytes, and fibroblasts. Processing and secretory pathways of specific proteases in mast cells were estimated based on morphological and immunohistochemical criteria, including exocytosis, secretion of individual granules, formation of macrovesicles, exosomes, and the mechanism of transgranulation [19,34].

Skin MCs without pathology most often had the Tr^+^Ch^+^CPA^3+^ protease phenotype (Figure 1a). Other phenotypes that lacked one of the specific proteases were less common. Histotopographically, it was possible to distinguish two subpopulations of MCs—located in close proximity to the epidermis in the reticular layer of the dermis (Figure 1b,c) and localized in deeper layers (Figure 1d–g). The former had the smallest dimensions, sometimes less than 10 µm. They were characterized by a small number of small granules in the cytoplasm of various protease profiles (Figure 1b,c). Often, the granules formed only a small perinuclear rim (Figure 1c). Sometimes, such cells were located in paracrine proximity to each other, forming groups.

Secretory granules differed from each other in the content of specific proteases; differences in immunopositivity to specific proteases of type I and type II secretory granules could be observed visually. At the final maturation of secretory granules, they reached sizes up to 0.4–0.5 μm, making it unfeasible to microscopically identify the peripheral localization of specific proteases typical for mature granules. Areas immunonegative to specific proteases antibodies were rarely detected in the center of such secretory granules; therefore, they occupied the entire area of the granules. MCs frequently had an elongated shape with the formation of cytoplasmic processes. The smallest MCs contained single tryptase-positive granules, with minimal content of other specific proteases.

Another subpopulation of skin MCs were larger cells located in the lower dermis, surrounding sebaceous glands, hair follicles, or included in small groups with fibroblasts and fibrocytes (Figure 1d–g). Such cell aggregates were often located between bundles of collagen fibers or near the microvasculature. MCs were not only larger in size, but were also filled with mature granules with sizes of 0.5 μm and more. The number of such granules and their location in the cytoplasm was quite variable; however, we noted complete filling of the cytoplasm in some cases, and more peripheral localization in others. Most often, the central part of the cytoplasm free from specific proteases was determined in these cases, despite the simultaneous intragranular localization of tryptase, chymase, and CPA3. Most secretory granules contained a triad of specific proteases. MCs were often located near fibroblasts or fibrocytes, and they most often showed secretory activity against specific proteases.

In melanoma development, histotopographic changes in mast cells were detected in the skin. First of all, there was a significant reduction, up to complete disappearance in some patients, of the subpopulation of small MCs located in close proximity to the epidermal basement membrane. The decrease in the number of small MCs under the epidermis is apparently associated with the intensification of their maturation and close involvement in the formation of the tumor microenvironment. The studied patients can be divided into two groups depending on the high or low mast cell infiltration of melanoma. In our study, we present data on patients manifesting high MCs content in the tumor microenvironment. In melanoma, diversification of the MC protease phenotype, which had individual features in each patient, was detected. Various options of intragranular colocalization of specific proteases appeared. In general, the prevalence of MCs combined with the simultaneous expression of three specific proteases resulted in the formation of new phenotypes in the population. Additionally, we observed MCs lacking one or two specific proteases, both tryptase and chymase, and CPA3 (Figure 2).

In melanoma, an increased frequency of MCs contacting with each other came to our attention (Figure 2a,b,d). Some contacting MCs had the same phenotypes of specific Tr^+^Ch^+^CPA3^+^ proteases (Figure 2a). However, there were frequent patterns of contact between MCs with different levels of proteases in some cells, in particular, without CPA3 (Figure 2b) or chymase (Figure 2c) expression. Colocalization of MCs with different phenotypes of specific proteases can be assumed as one of the mechanisms of intrapopulation interaction for the regulation of inductive properties in relation to the targets of the tumor microenvironment. Moreover, the apparent individuality of MCs in terms of the content of specific proteases, the content of which can vary significantly both in cells and in granules, introduces its own features into the final biological effects (Figure 2d).

The histotopographic features of intratumoral MCs consisted of several variants of colocalization in the tumor microenvironment of the skin and other localization: with tumor cells, immunocompetent cells, and stromal cells. In some patients, MCs were located both in the zone of paracrine effects on target cells and in direct contact with them. In most patients, MCs were located predominantly near tumor cells; less often, they were located near immunocompetent cells or fibroblasts (Figure 3, Figure 4 and Figure 5). However, contacts with leukocytes, in particular, with plasma cells dominated in other patients (Figure 5v,w). Histotopographic features of MCs depending on the phenotype of specific proteases in skin melanoma and other localizations have not been identified.

In tumor-associated MCs, crescentic formations were formed in the granules with a high level of specific proteases (Figure 5r,s), in contrast with the uniform peripheral distribution of proteases in secretory MC out of tumor zones.

This might relate to the mechanisms of intragranular protease redistribution due to changes in the biogenesis and secretory activities. In melanoma, MCs with an eccentric position of the nucleus (Figure 6a) or partially surrounded by granules were more common, which morphologically agreed with the MC polarization (Figure 6b,d,e). Moreover, MCs could transform into protease-positive post-cellular formations consisting only of the cytoplasm without visible signs of the nucleus (Figure 6c,f–h). In the tumor stroma, nucleus-free parts of the MC cytoplasm filled with protease-positive secretory material, including secretory granules, were more common (Figure 5 and Figure 6). These structures also had their own features in the content of specific proteases (Figure 6h), including patterns of intragranular distribution. There was an impression that these morphological features could be equivalent to the nucleus release from the MC cytoplasm, ultimately leading to the formation of large non-nuclear fragments of the cytoplasm. Regarding the mechanism for these structure formations, it can be assumed that under certain conditions, MCs are able to release almost the entire supply of secretory proteases, while maintaining the potential for increasing new synthesis. The following fact speaks for this point of view: there are morphological patterns of the nucleus release from the MCs (Figure 6b,d,e), which occur in response to increased demands in some areas of the secretome tumor microenvironment, including specific proteases. Large fragments of cytoplasm up to 10 µm in size, with cytoplasm filled with nucleus-free secretory granules, were quite common findings (Figure 6f–h). The preserved autonomy of such formations for a specific time is evidenced by secretory pathways realized via the release of specific proteases to tissue targets (Figure 5g,p and Figure 6c,f–h). In addition, the functioning of these formations while maintaining selective secretion and excretion of certain specific proteases is supported by the presence of granules with a different profile of chymase, tryptase, or CPA3 in these formations (Figure 6h).

There were new variants of directed secretion, which consisted of the formation of rather large MC processes oriented towards other cells in the tumor microenvironment. The biological meaning of this phenomenon may be the construction of temporary communications of MC with certain targets in the tumor microenvironment, which are the morphological basis for regulation (Figure 4e,f). Different variants of directed protease secretion to the region of a tumor cell or another target cell of the tumor microenvironment were represented in Figure 3a, Figure 4a,c–f and Figure 5. Most commonly, targeted secretion was accompanied by the tryptase and carboxypeptidase A3 release from the granules. Moreover, colocalization of some MCs with target cells, including tumor cells, seemed to be accompanied by the formation of a specialized contact zone over an area larger than 1 μm^2^ (Figure 3b and Figure 4b). In addition, the nuclei of tumor cells contacting with MCs might manifest immunopositivity for tryptase and CPA3 (Figure 5a–g). Sometimes, specific proteases were found in the nuclei of cells of the tumor microenvironment without the presence of MCs, which allows us to consider this phenomenon as quite long term compared to the migratory activity of MCs (Figure 5h–j).

In MCs, there was an intensification of the processes of biogenesis of secretory granules with their accumulation in the cytoplasm. The size variability of protease-positive secretory granules increased with their number. Due to tumor cells, MCs sometimes acquired a rather unusual shape, with abnormally large granule-like formations (Figure 3a). The intensification of the certain specific protease excretion was combined with more frequent directed secretion in the granule composition on target cells. One of the detected mechanisms for specific protease secretion in tumor-associated MCs was transgranulation (Figure 5a).

Features of specific protease secretion in the tumor microenvironment of skin melanoma were not only intensification if compared with normal skin MCs. The secretion selectivity was of great significance; this manifested itself in the diverse nature of the protease phenotype of secretory granules (Figure 7). The components of the secretome were often located in different granules during secretion. The physiological meaning of this event may lie in the selective effects of each of the specific proteases. It is necessary to highlight the need for separate secretion of specific proteases, not only to achieve certain effects of each enzyme (Figure 7a,b), but also to achieve the overall integral effect, which is a multi-stage development process over time.

The increased tryptase and carboxypeptidase A3 secretion on the targets of a specific tissue microenvironment should also be noted. We found that, first, virtually all tumor-associated MCs had a higher level of LAMP1 expression. LAMP1 localization was associated with a peripheral layer of granules in which specific proteases were confined (Figure 7e,f). A different degree of LAMP1 content was detected among MCs of the tumor microenvironment. In particular, it should be pointed out that MCs adjacent to tumor cells had the maximum expression of LAMP1 in granules.

Immunopositivity for CD63 was significantly higher in the MCs of patients of the control group (Figure 7c,d). In some cases, these were patterns of diffuse staining of MCs, with the exception of central regions of the secretory granules. Concomitantly, high immunopositivity to CD63 in MCs was detected along the periphery of the granules, confirming that mature granules could be a source of specific proteases during the formation of exosomes for extracellular regulation of tumor microenvironment targets (Figure 7c,d).

## 3. Discussion

MCs are unique representatives of a specific tissue microenvironment, closely involved in the formation of the tumor landscape due to participation in various mechanisms of immunogenesis. There is an opinion about the key role of innate immunity in the development, growth, and prognosis of malignant melanoma [35]. Normally, the largest number of MCs was in the surface layers of the skin, agreeing with the data of other studies [15,36]. The formation of melanoma was accompanied by a significant redistribution of the MC population with the depletion of the pool of superficial small-sized MCs.

In accordance with the content of proteases, MCs in the skin are representatives of the subpopulation expressing CPA3, tryptase, and chymase [37,38,39]. Many biomarkers have been developed in clinical practice to improve the diagnosis and prognosis of some neoplasms. Elevated tryptase levels are found in subgroups of patients with hematologic and solid cancers [40]. This was also demonstrated in our study on melanoma.

The development of melanoma led to an appearance of other protease phenotypes of tumor-associated MCs accompanied by two of the triad of specific proteases. Since this fact may be of separate diagnostic value, attention can be paid to further detailing polarization patterns of the protease phenotype depending on the prognosis. Consequently, we do not rule out that MCs may specialize into well-defined definitive forms that have a range of biological advantages over tumor-evolving effects.

The proportion of protease expression in each MC may be different at a particular point in time representing a dynamic value. Therefore, MCs can have an individual level of expression of each of the specific proteases with the same phenotypes, providing a fairly wide variability in the protease phenotype and functional potential. In particular, MCs with the Tr^+^Ch^−^CPA3^+^ immunophenotype can differ significantly from each other both in tryptase and CPA3 expression.

Our investigations have demonstrated cytotopographic and histotopographic features of MCs associated with melanoma. The variability in the number of MCs in the tumor may be related to the prognostic effects described in the previous studies. Obviously, the patterns of direct colocalization of MCs with tumor cells or their localization within the paracrine influence evidence the directed migration of mast cells to tissue targets. There is evidence of selective effects of MCs provided by cytokines, chemokines, growth factors, and other biologically active substances. In this study, we focused on the potential of the specific MC protease effects. The results obtained demonstrate that tumor-associated MCs have directed secretion of specific proteases to tumor cells; this secretion manifests personalized features in various patients. In the context of the protease effect specificity, we may support the view of other authors that tryptase is taken up by tumor cells [29]; tryptase-positive granules are released from mast cells and are widely distributed within the tumor tissue, suggesting that tryptase could impact the tumor microenvironment.

The detection of tryptase within the nuclei of mouse and human cells explores new biological properties of the enzyme in regulating the processes of implementing genetic information, gene expression, and controlling proliferative activity [31,32]. Our study has demonstrated that, along with tryptase, CPA3 transport also occurs. It can be assumed that this phenomenon is explained by a tryptase accompaniment with the formation of joint effects, or implementation of specialized functions of CPA3. Further studies are required to elucidate the results obtained.

We found a significant increase in the activity of biogenesis and secretory pathways of specific proteases in the tumor stroma. Directed secretion of three proteases simultaneously may have its own biological patterns, namely mechanisms of transport to molecular targets. It is possible that the triad of specific proteases ensures the greatest enzymatic preservation for one of them, which has direct effects on structural targets. We noticed that the secretion of tryptase and CPA3 primarily affected tumor cells. Our data are consistent with the findings of the studies, reporting that, in a number of malignancies, an increased serum tryptase level is detected due to both an increased tumor-associated population of tryptase-positive MCs and protease secretion [26,41]. It can be inferred that tryptase may play a protective role in melanoma or at an early stage of oncogenesis, and the enzyme detection in the serum may be a simple and useful biomarker to better understand tumor biology [41]. However, it should be pointed out that in the biomaterial of some patients, detection of MCs in melanoma was extremely rare. Presumably, the dynamics of the size of the tumor-associated MC population change during tumor development, challenging documental registration of the moment of the MCs’ greatest migration into the tumor microenvironment.

The results of triple multiplex immunolabeling demonstrated that the MC population in skin melanoma, as well as in melanoma of other localizations, has a heterogeneous protease profile. MC formation without one of the specific proteases is obviously not an organ-specific feature of their population, but is a targeted mechanism to achieve an increased number of MCs of a certain phenotype. MCs are probably able to rearrange the expression of specific protease genes depending on the conditions of cell localization in a specific tissue microenvironment. An increase in the MC pool with a certain phenotype in the population is part of the mechanism of tumor interaction with healthy tissues. MCs can directly affect tumor cells. Moreover, mechanisms for development of the effects are triggered not only by close proximity and cells having contact with each other, but also by paracrine influence, which is evidenced by the MC location at a 20–30-micron distance from the tumor target.

Of particular significance is colocalization of MC secretory products, namely granules, with tumor cells. As noted earlier, protease-positive secretory granules are often detected without the presence of MC, or at a considerable distance from it. This has also been demonstrated in laboratory animals [29]. Obviously, such patterns are formed during targeted granule exocytosis while maintaining further MC migration, when the granule as an autonomous subcellular formation is able to function in a specific way and exert selective biological effects. On the other hand, the presence of specific transport mechanisms that promote the purposeful migration of secretory granules after its secretion in the direction of strategic targets cannot be omitted. In this case, the granules containing specific proteases are able, firstly, to keep proteases in an active state and protect them from premature destruction. In addition, the granule is an effective mechanism for the transport of specific proteases over long distances, which allows biological activity to be maintained for a long time after secretion from the MC. We have noticed that in some patients, mast cells are able to exert effects simultaneously on several plasma cells. The participation of MCs in such interactions supports their immunocompetent function, a detailed understanding of which requires further study.

Interactions with fibroblasts are mediated effects on melanoma involving the stromal component, both in relation to the biogenesis of extracellular matrix components and the secretory function of fibroblasts in relation to growth factors and cytokines. It should be accepted as fact that in melanoma, there are loci with a high distribution density of MCs and zones with a complete absence of MCs. This raises other questions for further research related to the apparent influence of the tumor microenvironment on the mechanisms of MC chemotaxis.

In melanoma, we observed MC with the nucleus exit from the cytoplasm filled with granules. The biological meaning of this phenomenon is a debatable question, since, on the one hand, this mechanism may be part of the MC reaction to create extracellular reserves of specific proteases in the tumor microenvironment. Further, it is quite possible that such patterns are a presentation of the mass excretion of secretory granules in a certain locus, though not as a variant of anaphylactic degranulation, but as a variant of delayed secretome entry with the remaining selectivity of mediator excretion into the extracellular matrix. Similar to erythropoiesis, MC enucleation can be considered a phenotypic sign of terminal differentiation when the life cycle is completed. Secretory granules can maintain specific proteases in an active state for quite a long time and have biological effects during secretion. On the other hand, this phenomenon can be considered in the context of regulation of the MC population within the organ-specific tissue microenvironment, which occurs due to further mitosis of the nucleus leaving the cytoplasm, and thus an increase in the population. In addition, one cannot exclude the option of updating the program for the synthesis of specific components of the secretome under the inductive influence of a specific tissue microenvironment, during which secretory resources with the prevailed production of the necessary components will be created in MCs from a specific initial level.

Some studies have demonstrated accumulation of MCs at the border of skin tumors and maintenance of the disease progression and metastasis by inducing angiogenesis [42]. Angiogenic factors include heparin, tryptase, chymase, TGF-β, bFGF, and VEGF [43]. MCs release VEGF-C and VEGF-F, which support the process of lymphangiogenesis, when the lymphatic vessels act as a pathway for the spread of neoplastic cells [44]. Several studies have reported a better prognosis for patients who have mast cell infiltration of the tumor, suggesting protective effects. In our study, we have demonstrated that these effects may be mediated by specific proteases, including tryptase, chymase, and CPA3.

Previously, it has been hypothesized that polarization of the functional potential of MC into pro-oncogenic and antitumorigenic competencies may depend, among other things, on the level of tryptase expression [29]. Additionally, we assume that CPA3 exerts tumorigenic effect along with tryptase. The interaction of mast cells with tumor cells through tryptase and CPA3 leads to antitumorigenic effects, the mechanisms of which are subject to thorough analysis for further studies. Based on the data of multiplex immunodetection, we obtained facts about the MC migration to tumor cells, active secretion of specific proteases, and the development of direct and indirect antitumorigenic effects. Obviously, each of the specific proteases is involved, to a certain extent, in the development of molecular mechanisms. More details on the biological significance of chymase, tryptase and CPA3 can be found in the relevant reviews [17,18,19].

As morphological evidence of an increase in the secretory pathways of intratumoral MCs, one can consider intensification of the specific protease excretion into the extracellular matrix by means of exocytosis of individual secretory granules or protease-positive macrovesicles. Morphological evidence of intracytoplasmic fusion of granules with each other was more often observed in MCs colocalized with other cells of the tumor microenvironment. The high secretory activity of MCs resulted in the formation of protease-positive inductive zones that moved over considerable distances around the mast cell.

The detection of protease-positive MC granule secretion directly to the endothelium or the lumen of the microvasculature was an interesting finding, suggesting that this phenomenon may facilitate the spread of the inducing effect of proteases on tumor biological features, including the migration of leukocytes from the microvasculature into the tumor microenvironment (Figure 5o).

To discriminate antitumorogenic effects of specific proteases from the effects of a great amount of biologically active substances of the entire secretome is rather complicated; it has been observed that allergic reactions, in which massive degranulation of mast cells occurs, lead to regression of melanoma [45]. Significant antitumor effects were demonstrated in a study investigating possible effects of substances that enhanced mast cell degranulation. In particular, surfactin in the in vivo model of melanoma skin cancer and the B16F10 melanoma cell line [45] led not only to an increase in tumor infiltration by mast cells, but also to an increase in intratumoral histamine levels, levels of IgE, interferon gamma, interleukins (IL)-2, IL-6, IL-12, as well as tumor necrosis factor-α in serum. Notably, the activity of caspase-3, 8, and 9 and the activity of apoptosis of B16F10 cells increased in tumor cells. However, it can be obviously assumed that, along with the studied components of the secretome, specific proteases, which entered the extracellular matrix, also exerted their effects. It is also necessary to consider the components of the secretome with neuroendocrine effects, namely, histamine, VEGF, IL-8, heparin, and TGF-β [46].

Thus, a sufficient number of MCs expressing specific proteases in the tumor stroma apparently belong to a protective mechanism against tumor evasion from immunocompetent cells and can serve as a favorable prognosis for disease development and assessment of the effectiveness of ongoing therapy.

## 4. Materials and Methods

### 4.1. Case Selection

Skin biomaterial was obtained from five male and fourteen female patients with melanoma of the skin. The age range of the patients was 29–80 years. Every patient was in stage IV according to clinical information. The patients have not been subjected to immunotherapy. Three pathologists reviewed the samples independently. Only cases in which there was a unanimous agreement on the histological diagnosis of malignant melanoma were included in this study. No significant differences between the probes under study both in cytomorphology and immunophenotyping were found. Skin biomaterial from six patients without pathology (two male and four female patients) was used as controls.

This study was conducted in accordance with the principles of the World Medical Association Declaration of Helsinki “Ethical Principles for Medical Research Involving Human Subjects” and approved by the Institutional Review Board of the Institute for Hematopathology, Hamburg, Germany. The samples were retrieved from the files of the Institute for Hematopathology, Hamburg, Germany. Tissue samples were taken for diagnostic purposes. Informed consent was obtained from all subjects. The samples were qualified as redundant clinical specimens that had been de-identified and unlinked from patient information.

### 4.2. Tissue Probe Staining

Tissue probes left over during the routine diagnostic procedure were fixed in buffered 4% formaldehyde and routinely embedded in paraffin. Paraffin tissue sections (2 µm thick) were deparaffinized with xylene and rehydrated with graded ethanols according to a standard procedure [47].

### 4.3. Immunohistochemistry

For the immunohistochemical assay, we subjected deparaffinized sections to antigen retrieval by heating the sections in a steamer with R-UNIVERSAL Epitope Recovery Buffer (Aptum Biologics Ltd., Southampton, SO16 8AD, UK ), at 95 °C × 30 min. Blocking the endogenous Fc receptors prior to incubation with primary antibodies was omitted, according to our earlier recommendations [48]. After antigen retrieval and, when required, quenching endogenous peroxidase, sections were immunoreacted with primary antibodies. The list of primary antibodies used in this study is presented in Table 1. Immunohistochemical visualization of bound primary antibodies was performed either with Ventana Slide Stainer or manually, according to the standard protocol [47,49]. For manually performed immunostaining, primary antibodies were applied in concentration from 1 to 5 µg/mL and incubated overnight at +4 °C.

Bound primary antibodies were visualized using secondary antibodies (purchased from Dianova, Hamburg, Germany, and Molecular Probes, Darmstadt, Germany) conjugated with Cy3, Alexa Fluor-488, or Cy5. The final concentration of secondary antibodies was between 5 and 10 µg/mL PBS. Single and multiple immunofluorescence labeling were performed according to standard protocols [47]. The list of secondary antibodies and other reagents used in this study is presented in Table 2.

To simultaneously detect antigens from the same host species, we performed TSA with the subsequent heat elution treatment after each immunostaining step [50]. The bound primary/secondary antibody complex from the preceding immunolabeling step was thereby eluted with a citrate/acetate-based buffer, pH 6.0, containing 0.3% SDS (also available from VENTANA as CC2 solution, cat # 950-223) [51]. Nuclei were counterstained with 4’,6-diamidino-2-phenylindole (DAPI, 5 µg/mL in PBS) for 15 s, and the sections were then mounted using VectaShield (Vector Laboratories, Burlingame, CA, USA).

### 4.4. Controls

Control incubations were: omission of primary antibodies or substitution of primary antibodies by the same IgG species (Dianova, Hamburg, Germany) at the same final con-centration as the primary antibodies. The exclusion of either the primary or the secondary antibody from the immunohistochemical reaction and the substitution of primary antibodies with the corresponding IgG at the same final concentration resulted in a lack of immunostaining. The TSA step alone did not contribute to any specific immunostaining that might have influenced the analysis. Additionally, the specific and selective staining of different cells with the use of primary antibodies from the same species on the same preparation is, by itself, sufficient control for the immunostaining specificity.

### 4.5. Image Acquisition

Stained tissue sections were observed on a ZEISS Axio Imager.Z2 equipped with a Zeiss alpha Plan-Apochromat objective 100×/1.46 Oil DIC M27, a Zeiss Objective Plan-Apochromat 150×/1.35 Glyc DIC Corr M27 and a ZEISS Axiocam 712 color digital microscope camera. Captured images were processed with the software program “Zen 3.0 Light Microscopy Software Package”, “ZEN Module Bundle Intellesis & Analysis for Light Microscopy”, “ZEN Module Z Stack Hardware” (Carl Zeiss Vision, Jena, Germany) and submitted with the final revision of the manuscript at 300 DPI.

### 4.6. Data Availability

The authors declare that all the data supporting the findings of this work are available within the article or from the corresponding author upon reasonable request.

## 5. Conclusions

Our data imply that a sufficient number of MCs expressing specific proteases in the tumor stroma apparently are associated with a protective mechanism against tumor evasion from immunocompetent cells, and can serve as a favorable prognosis for disease development and assessment of the effectiveness of ongoing therapy.

Elucidating the molecular and morphological features of MC protease expression in melanoma allows us to look at the pathogenesis of the disease from a new perspective and can be used to clarify the MCs classification, the disease prognosis, and evaluate the effectiveness of ongoing antitumor therapy.

## Figures and Tables

**Figure 1 ijms-23-08930-f001:**
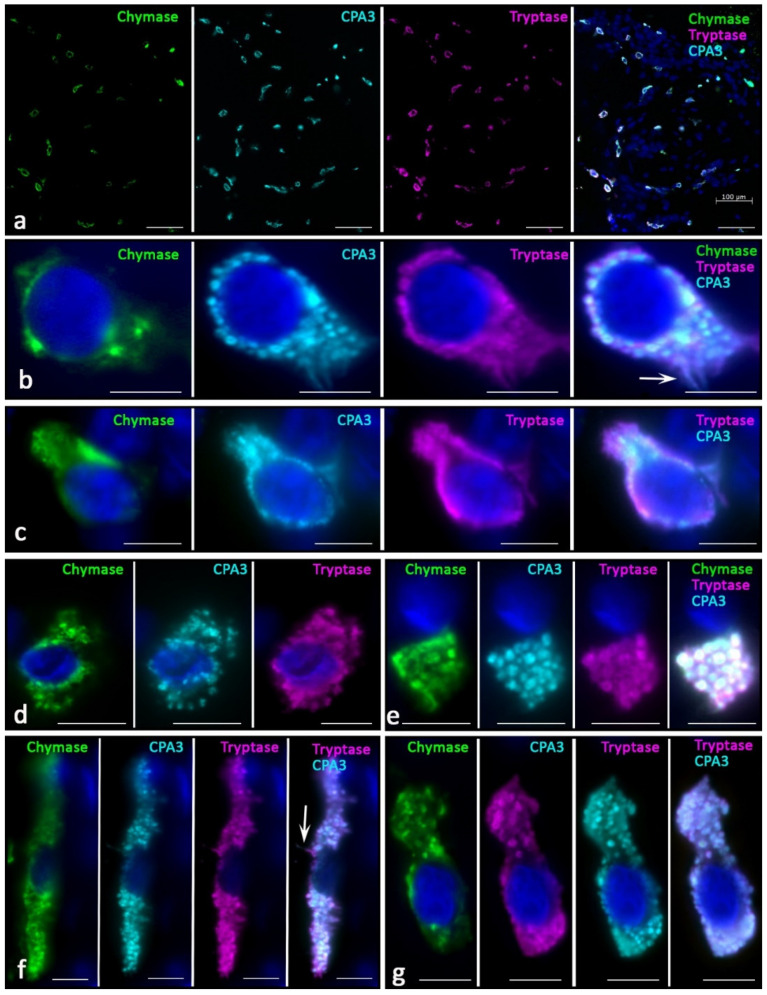
MC phenotype in normal skin. (**a**) General view of the MCs distribution in the skin, the vast majority of MCs contain a triad of specific proteases. (**b**,**c**) Representatives of the subepidermal subpopulation of the skin MCs, which are small in size and have a low content of specific proteases. Selective secretion of tryptase and carboxypeptidase A3 is detected (arrowed). (**d**–**g**) MCs localized in the deep layers of the dermis; (**d**)—MCs with the cytoplasm unevenly filled with secretory granules. (**e**) Fragments of the MC cytoplasm filled with protease-positive secretory granules. (**f**) Elongated MCs with selective secretion of tryptase and carboxypeptidase A3 (arrowed). (**g**) MCs with a triad of specific proteases, tryptase and carboxypeptidase A3 predominate. Scale bar: 100 µm (**a**), 5 µm (the rest).

**Figure 2 ijms-23-08930-f002:**
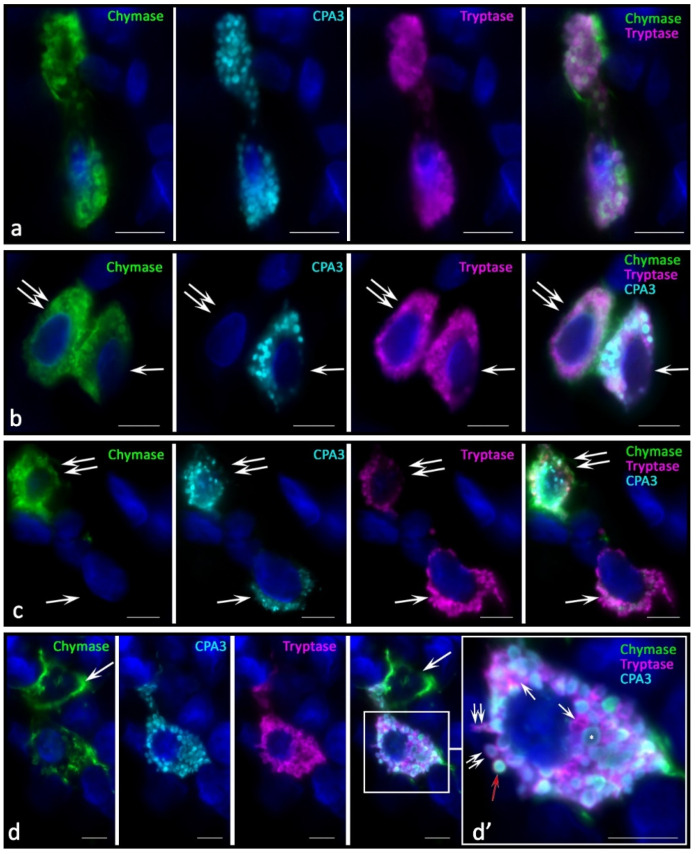
Phenotypes of specific proteases of MCs in the tumor microenvironment in melanoma. (**a**) Interaction of two MCs containing tryptase, chymase, and CPA3. (**b**) MC contacting with the triad of specific proteases expression (indicated by an arrow) with a CPA3-negative mast cell (indicated by a double arrow). (**c**) Chymase-negative MCs (indicated by an arrow) are co-localized within the limits of paracrine influence with MCs, which express a triad of specific proteases (indicated by a double arrow). (**d**) Chymase-positive MC contacting (indicated by an arrow) with a MC expressing tryptase, chymase, and CPA3. (**d’**) Fragment of the photo in (**d**). Granules with different content of specific proteases are visualized, including granules with the phenotype Tr^−^Ch^−^CPA3^+^ (indicated by an asterisk), Tr^+^Ch^−^CPA3^−^ (indicated by an arrow). Selective secretion of a dyad (“tryptase-CPA3”, double arrow) and a triad (red arrow) of specific proteases is detected simultaneously. Scale bar: 1 µm (**d’**), 5 µm (the rest).

**Figure 3 ijms-23-08930-f003:**
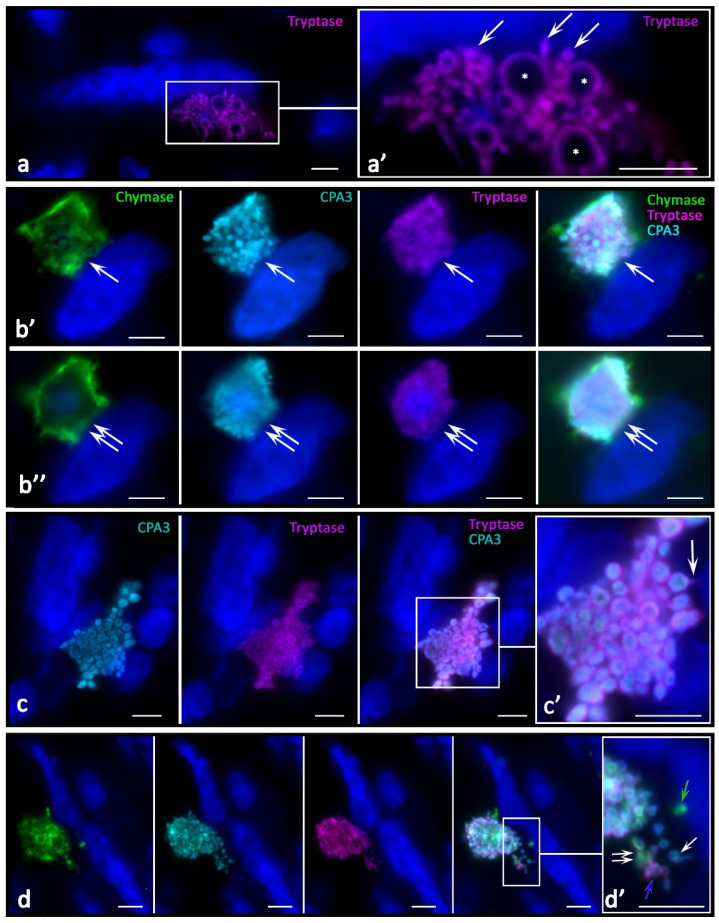
Direct interaction of mast cells with tumor cells through specific proteases. (**a**) Directed secretion of tryptase to the tumor cell (indicated by an arrow). The mast cell contains secretory granules of various sizes; some of them are abnormally large (indicated by an asterisk). Phenotype of a mast cell-tumor cell contact at visualization levels (**b’**,**b’’**); the distance between them along the Z axis is 1.5 µm (cell colocalization locus is indicated by an arrow at levels (**b’**) and double arrow at levels (**b’’**). (**c**) A large nucleus-free fragment of the MC cytoplasm, the secretory granules of which manifest the joint secretion of tryptase and CPA3 (indicated by an arrow). (**d**) Excretion of secretory granules by a mast cell with different phenotypes of specific proteases: chymase-positive (green arrow), with a predominant content of CPA3 (white single arrow), CPA3 and tryptase (blue arrow), as well as a triad of specific proteases (double arrow). Scale bar: 1 µm (**a’**,**c’**,**d’**), 5 µm (the rest).

**Figure 4 ijms-23-08930-f004:**
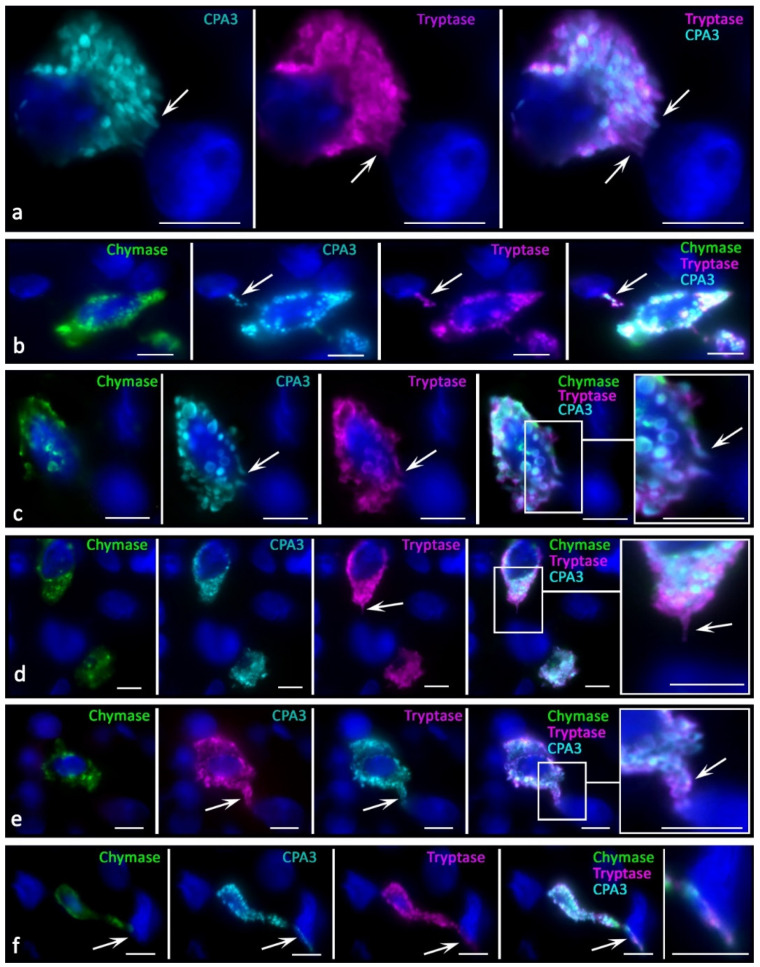
Targeted effect of specific proteases on cells of the tumor microenvironment in melanoma. (**a**) Combined intense targeted secretion of tryptase and CPA3 to an adjacent cell over a large area of influence (indicated by an arrow). (**b**) Release of secretory granules with the tryptase^+^CPA3^+^ phenotype; some of them reach the nucleus of the adjacent cell (indicated by an arrow). (**c**) Formation of a locus of tryptase and CPA3 directed secretion to a cell of the tumor microenvironment (indicated by an arrow). (**d**) Adjacency of two MCs to the target cell with selective exocytosis of tryptase into the nuclear region (indicated by an arrow). (**e**,**f**) Variants of the formation of the MC cytoplasm elongated section towards the target cells (indicated by an arrow). Scale bar: 5 µm.

**Figure 5 ijms-23-08930-f005:**
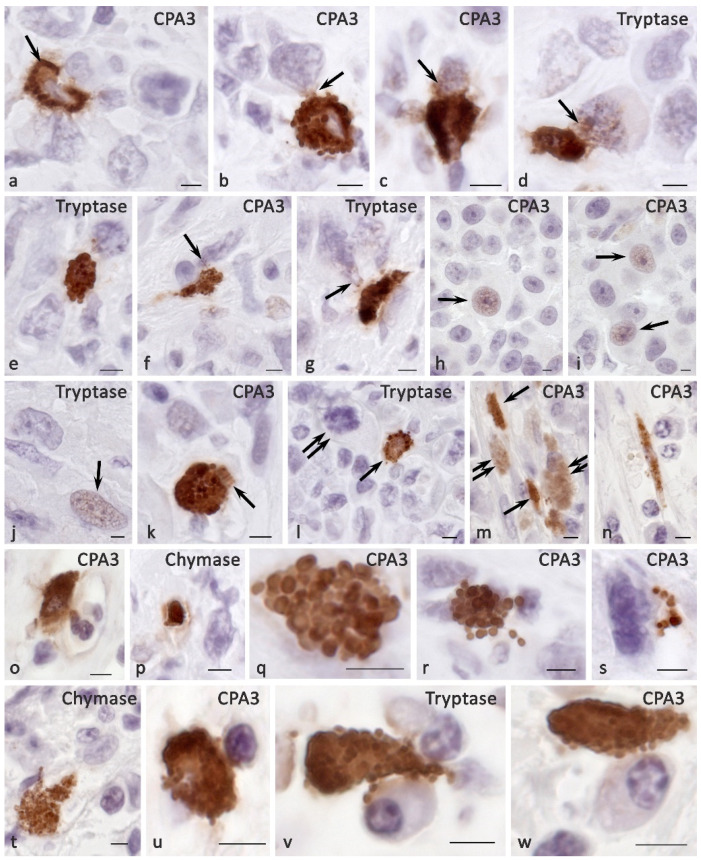
Histotopography and secretory pathways of specific MC proteases in the tumor microenvironment. (**a**) Transgranulation of CPA3 simultaneously towards several tumor cells. (**b**) Secretion of CPA3 in the composition of the granules to the tumor cell. (**c**–**g**) Variants of MC-specific protease secretion with signs of transfer into the nucleoplasm of tumor cells (indicated by an arrow). (**h**–**j**) Nuclei of tumor cells immunopositive to CPA3 (**h**,**i**) and tryptase (**j**) in the MC absence in the zone of paracrine effects (indicated by an arrow). (**k**) Massive secretion of CPA3 into the locus of the tumor microenvironment (indicated by an arrow). (**l**,**m**) Colocalization of MCs (indicated by an arrow) with tumor cells (indicated by a double arrow) within paracrine effects. (**n**,**o**) Adjacence of MCs with elements of the microvasculature. (**p**–**s**) Various options of the secretory activity of protease-containing fragments of the MC cytoplasm (**p**,**q**) and individual secretory granules (**r**,**s**). (**t**) Adjacence of MC with fibroblast. (**u**–**w**) Targeted secretion of specific MC proteases towards the plasmalemma of a lymphocyte (**u**) and plasmocytes (**v**,**w**). Scale bar: 5 µm.

**Figure 6 ijms-23-08930-f006:**
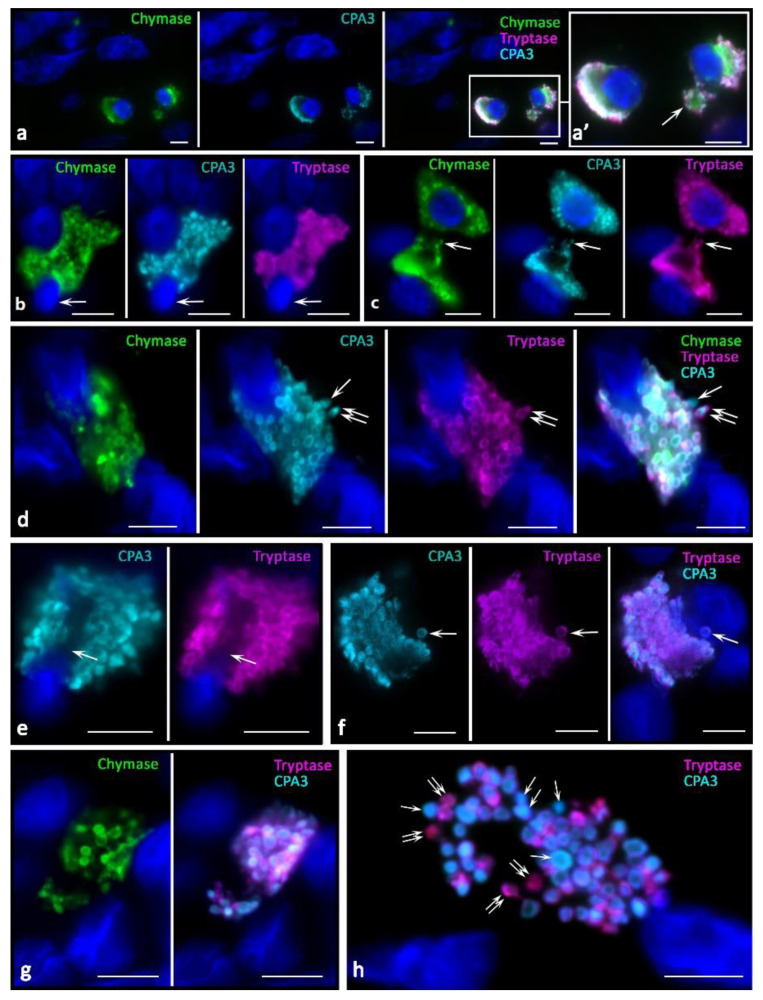
Protease phenotype and sectoral activity of MCs during the formation of macrovesicles and cytoplasts in the tumor microenvironment. (**a**) Eccentric position of the nucleus in MCs with the expression of a triad of specific proteases, with signs of detachment of a large cytoplasm fragment from one of them (indicated by an arrow). (**b**) Signs of the start of the nucleus release and MC enucleation (indicated by an arrow). (**c**) Colocalization of protease-positive MCs with a nuclear-free backbone of the cytoplasm with signs of secretory activity (indicated by an arrow). (**d**) Enucleation of the nucleus is accompanied by selective secretion as part of CPA3 granules (indicated by an arrow), or together with tryptase (indicated by a double arrow). (**e**) Completion of the nucleus exit from the MC with the formation of a cavity in the cytoplasm (indicated by an arrow). (**f**–**h**) The terminal stage of MC enucleation with the formation of a cytoplasmic backbone and preservation of the stock of specific proteases in the composition of the granules and secretory activity (**f**), indicated by an arrow). (**g**,**h**) Splitting the mother cytoplasm into two smaller fragments, the granules have a variability in the content of specific proteases, including granules with predominant accumulation of CPA3 (indicated by an asterisk) and tryptase (indicated by a double arrow). Scale bar: 5 µm.

**Figure 7 ijms-23-08930-f007:**
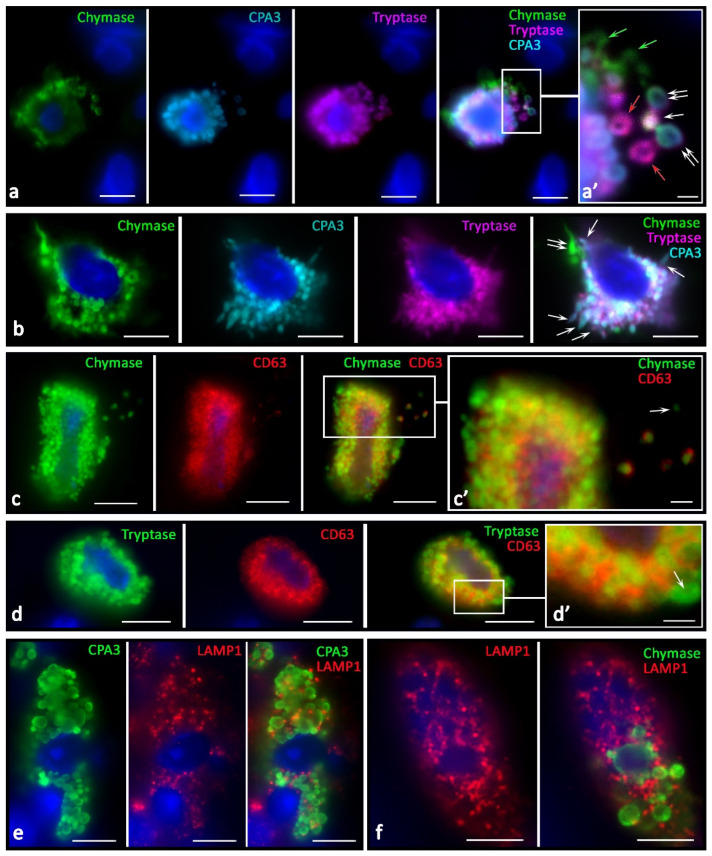
Morphological equivalents of secretory pathways of specific MC proteases in the tumor microenvironment. (**a**) Different phenotype of specific proteases of secretory granules in the intercellular substance after secretion: tryptase^+^ (red arrow), chymase^+^ (green arrow), tipase^+^chymase^+^ (white arrow), chymase^+^tryptase^+^CPA3^+^ (double arrow). (**a’**) The same as (**a**) at higher magnification. (**b**) Simultaneous secretion of tryptase combined with CPA3 (indicated by an arrow), and chymase (indicated by a double arrow). (**c**,**d**) High degree of immunopositivity to CD63 of chymase-containing and tryptase-containing MCs. Secretory granules differ in the degree of immunopositivity for CD63 in the extracellular matrix (**c’**) and MC cytoplasm (**d’**)**,** including the absence (indicated by the arrow). (**e**,**f**) LAMP1 expression in CPA3^+^ mast cell (**e**) and chymase^+^ MCs (**f**).

**Table 1 ijms-23-08930-t001:** Primary antibodies used in this study.

Antibodies	Host	Catalogue Nr.	Dilution	Sourse
Tryptase	Mouse monoclonal Ab	#ab2378	1:3000	AbCam, United Kingdom
Tryptase	Rabbit monoclonal Ab	#ab151757	1:2000	AbCam, United Kingdom
Chymase	Mouse monoclonal Ab	#ab2377	1:2000	AbCam, United Kingdom
Carboxypeptidases A1, A2 and B	Rabbit monoclonal Ab	#ab181146	1:500	AbCam, United Kingdom
Carboxypeptidase A3 (CPA3)	Rabbit polyclonal Ab	#ab251696	1:1000	AbCam, United Kingdom
CD63	Mouse monoclonal Ab	#ab1318	1:100	AbCam, United Kingdom
LAMP1	Rabbit polyclonal Ab	#ab24170	1 µg/mL	AbCam, United Kingdom

**Table 2 ijms-23-08930-t002:** Secondary antibodies and other reagents.

Antibodies and Other Reagents	Source	Dilution	Label
Goat anti-mouse IgG Ab (#ab97035)	AbCam, United Kingdom	1/500	Cy3
Goat anti-rabbit IgG Ab (#ab150077):	AbCam, United Kingdom	1/500	Alexa Fluor 488
AmpliStain™ anti-Mouse 1-Step HRP (#AS-M1-HRP)	SDT GmbH, Baesweiler, Germany	ready-to-use	HRP
AmpliStain™ anti-Rabbit 1-Step HRP (#AS-R1-HRP)	SDT GmbH, Baesweiler, Germany	ready-to-use	HRP
4’,6-diamidino-2-phenylindole (DAPI, #D9542-5MG)	Sigma, Hamburg, Germany	5 µg/mL	*w*/*o*
VECTASHIELD^®®^ Mounting Medium (#H-1000)	Vector Laboratories, Burlingame, CA, USA	ready-to-use	*w*/*o*
DAB Peroxidase Substrat Kit (#SK-4100)	Vector Laboratories, Burlingame, CA, USA	ready-to-use	DAB
Mayer’s hematoxylin (#MHS128)	Sigma-Aldrich	ready-to-use	*w*/*o*

## Data Availability

All data and materials are available on reasonable request. Address to I.B. (email: buchwalow@pathologie-hh.de) or M.T. (email: mtiemann@hp-hamburg.de) Institute for Hematopathology, Hamburg, Germany.

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
