# Peer review of "Protease Profile of Tumor-Associated Mast Cells in Melanoma"

_ijms, 2022, doi:10.3390/ijms23168930_

Round 1
Reviewer 1 Report
This is an interesting paper about protease profile of tumor-associated mast cells in melanoma.
However some minor changes are needed:
- In the Introduction, please add a little paragraph about melanoma (whatis is, epidemiology etc...).
- in the Discussion please speculate briefly about the association between mast cells and cancer and in this regard add this article in the references: Mast cells and cancer. G Ital Dermatol Venereol. 2019 Dec;154(6):650-668. doi: 10.23736/S0392-0488.17.05818-7. Epub 2017 Nov 30. PMID: 29192477.
- add also some sentence about the fact that mast cells may induce the onset of several melanocytic lesions. In this regard, read and add in the literature the follwing article Cutaneous mastocytosis combined with eruptive melanocytic nevi and melanoma. Coincidence or a linkage in the pathogenesis? J Dermatol Case Rep. 2014 Sep 30;8(3):70-4. doi: 10.3315/jdcr.2014.1179. PMID: 25324908; PMCID: PMC4195503.
- Finally, as also reported in recent article that you can speculate about adding also in your references (Serum tryptase levels in patients with psoriasis: a pilot study for a possible useful biomarker. Clin Exp Dermatol. 2022 Jan;47(1):178-179. doi: 10.1111/ced.14874. Epub 2021 Aug 30. PMID: 34363629.) mast cells may have an important role also in inflammatory disease (such as psoriasis) and this may have also important retracements in the future regarding the immune response to melanoma cells, above all in patients under immunotherapy.
Thank you.
Author Response
TO REVIEWER #1
The Authors (A) thank the Reviewer (R) for the analysis of the study, the positive assessment of the results obtained, and the comments made.
R.- In the Introduction, please add a little paragraph about melanoma (whatis is, epidemiology etc...).
A.- We have added in the Introduction the following paragraph: “Melanoma is the most fatal skin cancer. As a result, melanoma continues to be the subject of several preclinical and clinical investigations to further understand cancer immunobiology [1]. Human and animal studies addressing potential functions of mast cells (MCs) in skin cancers have provided conflicting results. Many unanswered questions need to be addressed before we understand whether tumor-associated MC are adversaries, allies or simply innocent bystanders in different types and subtypes of skin cancers [2]”.
R.- in the Discussion please speculate briefly about the association between mast cells and cancer and in this regard add this article in the references: Mast cells and cancer. G Ital Dermatol Venereol. 2019 Dec;154(6):650-668. doi: 10.23736/S0392-0488.17.05818-7. Epub 2017 Nov 30. PMID: 29192477.
A.- We have added in the Discussion the following paragraph: “In accordance with the content of proteases, MCs in the skin are representatives of subpopulation expressing CPA3, tryptase and chymase [1-3]. Many biomarkers have been developed in clinical practice for improving diagnosis and prognosis of some neoplasms. Elevated tryptase levels are found in subgroups of patients with haematologic and solid cancers [4]. This was also demonstrated in our study on the melanoma.”
R.- add also some sentence about the fact that mast cells may induce the onset of several melanocytic lesions. In this regard, read and add in the literature the following article Cutaneous mastocytosis combined with eruptive melanocytic nevi and melanoma. Coincidence or a linkage in the pathogenesis? J Dermatol Case Rep. 2014 Sep 30;8(3):70-4. doi: 10.3315/jdcr.2014.1179. PMID: 25324908; PMCID: PMC4195503.
A.- We have added in the Introduction the following sentence: “A possible pathogenetic linkage between cutaneous mastocytosis and melanoma is being actively discussed [5]; MCs appear to be involved either in anti-tumor processes or, conversely, in the mechanisms of oncogenesis stimulation.”
R.- Finally, as also reported in recent article that you can speculate about adding also in your references (Serum tryptase levels in patients with psoriasis: a pilot study for a possible useful biomarker. Clin Exp Dermatol. 2022 Jan;47(1):178-179. doi: 10.1111/ced.14874. Epub 2021 Aug 30. PMID: 34363629.) mast cells may have an important role also in inflammatory disease (such as psoriasis) and this may have also important retracements in the future regarding the immune response to melanoma cells, above all in patients under immunotherapy.
A.- We have added in the Introduction the following sentence: “MCs may have an important role also in inflammatory disease (such as psoriasis) and this may have also important retracements in the future regarding the immune response to melanoma cells, above all in patients under immunotherapy [6].
Thank you.
Igor Buchwalow
REFERENCES
- Elieh Ali Komi, D.; Kuebler, W.M. Significance of Mast Cell Formed Extracellular Traps in Microbial Defense. Clin Rev Allergy Immunol 2021, doi:10.1007/s12016-021-08861-6.
- Tebroke, J.; Lieverse, J.E.; Safholm, J.; Schulte, G.; Nilsson, G.; Ronnberg, E. Wnt-3a Induces Cytokine Release in Human Mast Cells. Cells 2019, 8, doi:10.3390/cells8111372.
- Gurish, M.F.; Austen, K.F. Developmental origin and functional specialization of mast cell subsets. Immunity 2012, 37, 25-33, doi:10.1016/j.immuni.2012.07.003.
- Paolino, G.; Corsetti, P.; Moliterni, E.; Corsetti, S.; Didona, D.; Albanesi, M.; Mattozzi, C.; Lido, P.; Calvieri, S. Mast cells and cancer. G Ital Dermatol Venereol 2019, 154, 650-668, doi:10.23736/s0392-0488.17.05818-7.
- Donati, P.; Paolino, G.; Donati, M.; Panetta, C. Cutaneous mastocytosis combined with eruptive melanocytic nevi and melanoma. Coincidence or a linkage in the pathogenesis? J Dermatol Case Rep 2014, 8, 70-74, doi:10.3315/jdcr.2014.1179.
- Paolino, G.; Di Nicola, M.R.; Currado, M.; Brianti, P.; Mercuri, S.R. Serum tryptase levels in patients with psoriasis: a pilot study for a possible useful biomarker. Clin Exp Dermatol 2022, 47, 178-179, doi:10.1111/ced.14874.
Reviewer 2 Report
Interesting study on mast cells in melanoma.
Comments:
- no clinical correlation of mast cells and overall patient outcome is provided; do the authors have survival data for those patients?
- the numbers of the control patients does not add up: 2+4 is 6 and not 5; please explain
- There are beautiful pictures provided for histopathology and immunohistochemistry; however no quantification of findings has been provided. It would be helpful if the authors could provide an objective quantification comparing melanoma versus controls
Author Response
TO REVIEWER #2
The Authors (A) thank the Reviewer (R) for the analysis of the study, the positive assessment of the results obtained, and the comments made.
(R) - no clinical correlation of mast cells and overall patient outcome is provided; do the authors have survival data for those patients?
(A) - The biomaterial was obtained from patients recently, so we are unable to assess the survival data for those patients. Given the importance of this issue for the study, we relied on the data of other authors, in particular, “Siiskonen, H.; Poukka, M.; Bykachev, A.; Tyynela-Korhonen, K.; Sironen, R.; Pasonen-Seppanen, S.; Harvima, I.T. Low numbers of tryptase+ and chymase+ mast cells associated with reduced survival and advanced tumor stage in melanoma. Melanoma Res 2015, 25, 479-485, doi:10.1097/CMR.0000000000000192.» and other authors.
(R) - the numbers of the control patients does not add up: 2+4 is 6 and not 5; please explain.
(A) – Thank you. We have corrected it correspondingly: “Skin biomaterial from 6 patients”.
(R) - There are beautiful pictures provided for histopathology and immunohistochemistry; however no quantification of findings has been provided. It would be helpful if the authors could provide an objective quantification comparing melanoma versus controls
(A) - This study has some limitations, such as the low sample numbers, which prevented us performing statistically objective quantification analysis. However, the patterns of histotopographic features of tumor-associated MCs and their integration into the immune landscape of the tumor microenvironment was objectively presented in this study in the form of descriptive data.
We hope that, as the study continues and the number of patients increases, we will also be able to conduct an objective quantification comparing melanoma versus normal controls. But this will be a new separate independent study.
Thank you.
Igor Buchwalow